# Clinical Significance of ROMs, OXY, SHp and HMGB-1 in Canine Leishmaniosis

**DOI:** 10.3390/ani11030754

**Published:** 2021-03-09

**Authors:** Michela Pugliese, Alessandra Sfacteria, Gaetano Oliva, Annastella Falcone, Manuela Gizzarelli, Annamaria Passantino

**Affiliations:** 1Department of Veterinary Sciences, University of Messina, 98168 Messina, Italy; michela.pugliese@unime.it (M.P.); alessandra.sfacteria@unime.it (A.S.); annastella.falcone@unime.it (A.F.); annamaria.passantino@unime.it (A.P.); 2Department of Veterinary Medicine and Animal Production, University of Naples Federico II, 80137 Naples, Italy; gaeoliva@unina.it

**Keywords:** canine leishmaniosis, HGMB-1, ROMs, OXY, SHp, OSi

## Abstract

**Simple Summary:**

The reactive oxidative metabolites (ROMs), the antioxidant barrier (OXY), the ratio between ROMs and OXY (OSi), thiol groups of plasma compounds (SHp) and the concentration of High Mobility Group Box-1 protein (HMGB-1) were evaluated in dogs naturally infected by *Leishmania* spp., correlating them with hematological and biochemical parameters. Results showed a significant increase in SHp levels, and a negative correlation between ROMs and the number of red blood cells, and between HGMB-1 and SHp, suggesting the potential role of oxidative stress in the pathogenesis of canine leishmaniosis.

**Abstract:**

This study aimed to investigate the role of oxidative stress parameters (ROMs, OXY, SHp), the Oxidative Stress index (OSi), and High Mobility Group Box-1 protein (HMGB-1) in canine leishmaniosis (CanL). For this study, thirty dogs, naturally infected with *Leishmania* spp. (Leishmania Group, LEISH) and ten healthy adult dogs (control group, CTR) were included. The diagnosis of CanL was performed by a cytological examination of lymph nodes, real time polymerase chain reaction on biological tissues (lymph nodes and whole blood), and an immunofluorescence antibody test (IFAT) for the detection of anti-*Leishmania* antibodies associated with clinical signs such as dermatitis, lymphadenopathy, onychogryphosis, weight loss, cachexia, lameness, conjunctivitis, epistaxis, and hepatosplenomegaly. The HMGB-1 and oxidative stress parameters of the LEISH Group were compared with the values recorded in the CTR group (Mann Whitney Test, *p* < 0.05). Spearman rank correlation was applied to evaluate the correlation between the HMGB-1, oxidative stress biomarkers, hematological and biochemical parameters in the LEISH Group. Results showed statistically significant higher values of SHp in the LEISH Group. Specific correlation between the ROMs and the number of red blood cells, and between HGMB-1 and SHp were recorded. These preliminary data may suggest the potential role of oxidative stress in the pathogenesis of CanL. Further studies are undoubtedly required to evaluate the direct correlation between inflammation parameters with the different stages of CanL. Similarly, further research should investigate the role of ROMs in the onset of anemia.

## 1. Introduction

The Canine leishmaniasis (CanL), caused by *Leishmania infantum*, is a vector-borne zoonotic protozoan disease transmitted by the bite of a female phlebotomine sand fly. *Leishmania infantum* is the same causative agent of the Zoonotic Visceral Leishmaniasis (ZVL), a disease endemic in the region of the Mediterranean basin, Africa, Asia, and South America, affecting more than 100,000 people annually [1,2]. The dog is considered the main reservoir of the parasite and plays an important role in the transmission of the disease to humans [3].

Alterations of the immune system and the inflammatory response are widely documented [4,5,6]. The disease is characterized by an exacerbated humoral immune response with a related depression of cellular immune response against the parasite [7,8,9,10]. The appearance of a broad spectrum of clinical signs (cutaneous alterations, lymphadenomegaly, onychogryphosis) and/or pathological abnormalities such as glomerulonephritis, polyarthritis, or uveitis derived from immune-complexes deposition [11,12,13] are considered the consequence of the immunological imbalance.

The severity of the disease is strongly related to the immune reaction of the host [4,5,6,14]. *Leishmania* spp. are able to evade the immune system and perpetuate the infection through the inhibition of the inflammatory cell oxidative metabolism and by using macrophages and neutrophils as carriers during the early stages of infection [15,16,17,18]. Once the disease is established, neutrophils restore peroxide production and contribute to the oxidative stress [19,20].

It has been suggested that oxidative stress may play an important role in the pathogenesis of hepatic and renal damage during *Leishmania* spp. infection [21,22,23].

High Mobility Group Box-1 protein (HMGB-1) is a DNA-associated nuclear protein secreted by activated macrophages and neutrophils during the inflammation and it passively leaks from necrotic or damaged cells [24,25]. HMGB-1 released into the intravascular space acts as an alarmin with the ability to intensify local inflammatory responses, interacts with endothelial cells, and activates the secretion of soluble pro-inflammatory mediators [26,27,28].

Some authors have highlighted that the High Mobility Group Box-1 protein (HGMB-1) is integral to the response to oxidative stress [27]. The release of damage-associated molecular patterns (DAMPs), including HGMB-1, is reported as the result of cellular inflammation induced by *Leishmania* infection [28].

In addition, HMGB-1 is reported as a marker of various canine diseases including systemic inflammatory response syndrome, lymphoma, prostate cancer, and babesiosis [29,30,31,32].

The present study aimed to evaluate the reactive oxidative metabolites (ROMs), the antioxidant barrier (OXY), the ratio between ROMs and OXY (OSi), thiol groups of plasma compounds (SHp) and the concentration of HGMB-1 in blood samples of dogs naturally infected with *Leishmania* spp. Moreover, the objective was to investigate the relationship between these variables and the hematological and biochemical parameters of liver and/or kidney injuries.

## 2. Materials and Methods

### 2.1. Ethical Statement

The study was conducted according to the Italian Laws (L. n. 281/1991, L.R. 15/2000, Dlgs. 26/2014) and the standards recommended by the European Council Directive 2010/63/EU. It was approved by the Ethics Committee of the Department of Veterinary Sciences, University of Messina (Approval number: 22; date of approval 9 June 2018). Dogs enrolled in the study were recruited at a shelter belonging to an animal protection organization in Messina (Sicily, South Italy). Dog Shelters’ Administration was informed of the research objectives and the clinical procedures. Before sample and data collection, a signed informed consent form was obtained.

### 2.2. Animals

A total of 40 neutered mixed-breed dogs of both genders housed in a shelter were included in the study.

There were 30 dogs (19 males and 11 females, with an average body weight of 17.5 kg ± 2.8 and a mean age 5.1 ± 2.2 years old) naturally infected with *L. infantum* (Group LEISH), while the other 10 healthy dogs (7 males and 3 females, with a bodyweight of 14.1 kg ± 2.7 kg, and a mean age of 6.4 ± 3.7 years old), who were serologically, parasitologically and molecularly negative, were considered as the control group (Group CTR).

Dogs with associated conditions, such as neoplastic, endocrine, metabolic, infectious (*Ehrlichia*, *Anaplasma* spp.), and parasitic (*Babesia* spp., microfilariae) diseases, were excluded from the study.

All animals did not receive any drug therapy in the last 30 days before the onset of the sampling. The diagnosis of CanL was made by the detection of amastigotes in the cytological examination of lymph nodes aspirates by the immunofluorescence antibody test (IFAT), and by a real time polymerase chain reaction (PCR) [3,8,33].

Sera were tested for the presence of antibodies against *L. infantum* by IFAT using MHOM/TN/80/IPT1 as a whole parasite antigen preparation fixed on multispot slides provided by CReNaL (National Reference Center for Leishmaniosis, Istituto Zooprofilattico Sperimentale della Sicilia, Palermo, Italy), and a fluorescent labeled anti-canine gamma globulin (Sigma Aldrich, St. Louis, MO, USA) was used as the conjugate. The cut-off value was established at 1:80. The reading was performed by the same observer (AP) using a fluorescence microscope. Positive and negative controls provided by CReNaL were added for each test to verify the validity of the results [34].

*Leishmania* real-time polymerase chain reaction (PCR) was performed on whole blood and lymph node samples using the routine diagnostic services, primers and methods already validated [35].

All infected dogs were classified into four different clinical stages according to the presence of clinical signs and laboratory results, as proposed by the LeishVet group [11,36].

### 2.3. Collection of Blood Samples

Blood samples were collected at the same time (9:00 a.m.) by jugular or cephalic venepuncture from each dog using the vacutainer blood collection system. Blood was drawn into two different types of tubes: one containing tripotassium ethylenediaminetetraacetic acid (K_3_EDTA) (S-Monovette^®^ Sarstedt, Nümbrecht, Germany) for complete blood count (CBC) and the other with no additive. After clotting and centrifugation (ALC 4235 A, Milan, Italy) at 3000× g for 20 min at room temperature, the obtained serum was transferred in Eppendorf tubes with caps and stored at −80 °C until the analysis.

### 2.4. Samples Analysis

ROMs were assessed by the levels of hydroperoxides (R-ooH) generated by the peroxidation of lipids, amino acids, proteins, and nucleic acids during tissue damage and the molecules photometrically detected following the t reaction with a properly buffered chromogen. Values directly related to the color intensity were expressed in Carratelli units (1 CaRR u = 0.08 mg% hydrogen peroxide).

OXY was evaluated by the assay of residual unreacted radicals present after the oxidant action of an excess of hypochlorous acid with the plasma within the water solution. A decrease in values is directly correlated with the alteration of the plasma barrier due to oxidation and consequently to the severity of the injury. OXY values were expressed in mmol/L.

The ratio between the values of derivatives of reactive oxygen metabolites (d-ROMs) and OXY (×100) (Oxidative Stress index, OSi) is an arbitrary value, used as an index of the plasma redox status; high values indicate a higher concentration of oxidized molecules than non-enzymatic antioxidants [37].

SHp were assayed by the ability of thiol groups to develop a colored complex when reacting with DTNB (5,5-dithiobis-2-nitrobenzoic acid). Decreased values are directly correlated with a lower efficacy of the thiol antioxidant barrier.

HMGB-1 was measured in EDTA plasma by means of a commercially available human ELISA kit (IBL-International, Hamburg, Germany) previously validated for canine species [38].

Oxidative stress parameters (ROMs, OXY, and SHp) in the serum samples were assessed with an ultraviolet spectrophotometer (Slim SeaC, Florence, Italy) with a “spin traps” technique, based on the detection of colored derivatives complexes coming from the reaction between oxygen metabolites and free radicals.

Hematological examination including red blood cell count (RBC), hematocrit (PCV), hemoglobin concentration (Hgb), blood platelet count (PLT), and differential white blood cell count (WBC) was performed on each K_3_EDTA sample by a hematology analyzer (Procyte Dx, IDEXX Laboratories, Inc., Westbrook, ME0).

The serum obtained from centrifugation was used to perform a biochemical examination using a chemistry analyzer (Catalyst Dx, IDEXX Laboratories, Inc., Westbrook, ME, USA). Blood urea nitrogen (BUN), creatinine (CREA), total protein (TP), alanine-amino transferase (ALT), aspartate-amino transferase (AST), albumin (ALB) and globulin (GLOB) were assessed. All samples were assayed in duplicate.

### 2.5. Statistical Analysis

The statistical analysis was performed using SPSS for Windows package (Version 17.0, SPSS, Inc., Chicago, IL, USA). All results were expressed as the mean ± standard deviation (SD). Distribution of the data was tested by the Kolmorov-Smirnov test. An unpaired t-test was performed to assess the differences between the two groups. For the determination of the relationship between the oxidative stress markers and hematological and biochemical variables, Spearman’s rank test was used. The significance level was set at *p* = 0.05.

## 3. Results

Eleven dogs (11/30; 33%) belonging to LEISH group were classified as Leishvet stage I, nine (9/30; 27%) were in stage II, seven (7/30; 21%) in stage III, and three (3/30; 9%) in stage IV [11]. Clinical findings included cutaneous lesions (onychogryphosis, ulcerations, erythema, alopecia, popular dermatitis, exfoliative dermatitis), ocular diseases (blepharitis, keratitis, uveitis), peripheral lymph-adenomegaly, weight loss, and epistaxis (Table 1). Changes in the laboratory analytes included anemia, leucocytosis, thrombocytopenia, hyperglobulinemia, hypoalbuminemia, an increase of BUN, and CREA.

The values of each variable expressed as a mean and SD are reported in Table 2 and Table 3. The dogs belonging to the LEISH group showed a statistically higher value of SHp compared to the control group (*p* = 0.047, 171.63 ± 134.2 μmol/L versus 65.59 ± 18.26 μmol/L). No significant differences were observed in values of ROMs, OSi, and HGMB-1 (Table 2). Although SHp levels were above reference ranges [39] for the species examined, they were significantly higher (*p* = 0.032) in dogs belonging to Group LEISH clinically staged as I–II (154 ± 140 μmol/L) compared with dogs staged as III–IV (82.70 ± 67.68 μmol/L). A statistically significant difference in the number of red blood cells (RBC) (*p* = 0.13), hemoglobin (Hgb) (*p* = 0.17), and packed cell volume (PCV) (*p* = 0.05) concentration was observed between the two groups. Statistically significant higher values of BUN (*p* = 0.02) and creatinine (CREA) (*p* = 0.001) were detected in Group LEISH than in CTR (Table 3).

Spearman’s correlation and regression (R^2^) analysis of the data obtained from the LEISH group showed a negative correlation between SHp and HGMB-1 (*p* < 0.001). A positive correlation was detected between OSi and ROMs (*p* = 0.03; r = 0.77). A negative correlation was recorded between ROMs and RBC (*p* = 0.02; r = −0.57), OSi and OXY (*p* < 0.001; r = −0.79), and OSi and TP (*p* = 0.01; r = −0.58). No correlation was found between hematological and biochemical parameters. Data are reported in Table 4 and Table 5.

## 4. Discussion

This study aimed to evaluate the possible changes in the oxidative stress biomarkers and HGMB-1 in dogs affected by CanL and to correlate these values with hematological and biochemical parameters. The inflammatory response is addressed to impair the action of pathogens and it is exploited throughout the production and release of several mediators. Amongst them, oxidant compounds are released from neutrophils and macrophages after cytokines and phagocytosis stimulation. Oxidative stress is considered to have an important role in diseases and aging, as the cause and/or contributing factor [42]; its potential as a prognostic biomarker of non-resolving inflammation is under evaluation both in human and veterinary medicine. Oxidative stress results from an imbalance between the production of reactive oxygen species and the capacity of the cells to readily detoxify the injurious oxidants or easily repair the resulting damage [43]. The utility of oxidative stress biomarkers could be particularly important in CanL since it has been proved that in the early phase of infection, the *Leishmania* parasite inhibits the phagocytes production of oxidants, survives, and replicates within macrophages [22]. After this period of latency, inflammation takes advantage and phagocytes, especially neutrophils, burst the production of oxygen radicals as a defense mechanism against the host [9,44].

The possible role of any oxygen-free radical in CanL has been investigated in the recent years, showing its involvement in the pathogenesis of the disease [9,45,46,47].

Although the mechanisms for regulating HMGB-1 release and activity are not clear, experimental studies indicate that oxidative stress is a common mechanism regulating HMGB-1 translocation, release, and activity [24]. One major contributor to oxidative damage is H_2_O_2_, which is converted from superoxide that leaks from the mitochondria. H_2_O_2_ induces both active and passive HMGB-1 release from macrophage and monocyte cultures in a time- and dose-dependent fashion [48]. HGMB-1 plays an important role in the pathogenesis of inflammation-associated diseases [49,50].

In this study, significantly higher values of SHp were observed in the LEISH Group in stages I–II of disease. Although SHp is a component of oxidative stress like ROMs and OXY, its increase suggests that SHp values could be a more sensitive biomarker [47] in the clinically moderate stages of CanL, since their activation by *Leishmania* spp. switch off the cytokine and LPS activation of the macrophage respiratory burst above all the NO production, thus allowing the parasite to survive [51].

Results have not shown a significant increase in HGMB-1 in dogs affected by CanL in different stages but have indeed demonstrated a negative correlation with SHp values. This correlation could be strongly and negatively dependent on the oxidative stress-species activity that inhibits the cellular release of HGMB-1 [52].

Our results showed significantly higher OSi values in the LEISH than the CTR group. OSi allows a global view of the degree of oxidative stress because it relates to the pro and antioxidant status, thus avoiding data misinterpretation. It is well known that high values of OSi could reflect an imbalance between oxidant and antioxidant systems, with an over-accumulation of ROMs which can affect membrane structure and, consequently, changes in their fluidity [53,54,55]. A significant positive correlation was detected between OSi and ROMs, while a negative correlation was observed between ROMs and RBC in the LEISH group.

The oxidative part is the measurement of total peroxides or hydroperoxides (ROMs), produced from the oxidation of numerous molecules, such as lipids, proteins, and amino acids; therefore, this is a measure of oxidative impairment.

As the ideal balance between oxidants and antioxidants, the positive correlation between OSi and ROMs detected in the present study should indicate a predominant oxidant status in dogs affected by CanL, resulting, presumably, from the host defence mechanism against the increase in oxidation caused by the parasite [45]. As suggested from previous studies, the degree of oxidative damage may be related to the clinical stages of disease [56,57,58].

In the present study, the dogs with leishmaniosis showed a significantly lower value of RBCs, Hgb, and PCV, indicating the presence of anemia. Anemia is clinically defined as decreased hemoglobin and hematocrit levels, and/or a reduced erythrocyte count. A negative correlation was found between the OSi and the erythrocyte count.

Anemia is one of the most common laboratory findings reported in dogs affected by CanL [59,60], but its pathogenesis is complex and not definitively assessed.

Different organs may suffer severe hypoxia, as determined by the changes of these blood parameters [61]. This blood finding may be related to bleeding, hemolysis, inflammation, renal failure, chronic disease, and marrow aplasia or hypoplasia [62]. A shortened lifespan of RBCs associated with a modification in membrane lipid fluidity following oxidative stress has been reported in dogs with CanL [45,63].

The negative correlation between the OSi and erythrocyte count, reported in the present study, suggests the critical role of ROMs in the development of anemia.

ROMs are strongly correlated with anemia and play a crucial role in eryptosis. Indeed, oxidative stress may induce an early and altered eryptosis, not compensated by adequate erythropoiesis [64,65].

The increase of oxidative intra- and extra-erythrocytic stress causes erythrocyte membrane damage and deformity, causing accelerated hemolysis [61]. Although oxidative stress is not the primary etiology of these different types of anemia, it mediates several mechanisms involved in their onset [66].

With regard to the white blood cell count, hypoalbuminemia, hyperglobulinemia, and an increase in urea and creatinine are common laboratory findings described in CanL [67,68]. According to previous reports, the renal damage is attributed to immune complex glomerulonephritis and tubulointerstitial nephritis, undoubtedly considered the main cause of death in the affected animals [69,70]. Although no significant correlation appeared, uremia may be considered a contributor to the onset of anemia by the toxic effects on red blood cells, decreasing the half-life of erythrocytes [71,72].

## 5. Conclusions

Although oxidative stress has been reported in humans and dogs [56,57,58,73,74] with *Leishmania* infection, the role of HGMB-1 in this parasitic disease remains understood. The present study shows that leishmaniosis promotes clear alterations in oxidative stress and hematological findings. Even in a small cohort of cases, we pose the attention to the value of OSi and SHP as useful markers of stress in the clinically moderate stages of *Leishmania* infection. Further research is required to evaluate the correlation between parasite density, clinical status, and oxidative stress markers in dogs with leishmaniosis.

## Figures and Tables

**Table 1 animals-11-00754-t001:** Clinical staging of dogs affected by CanL (LEISH Group).

Clinical Stage	*n* (sex)	Serology	PCR/Cytology	Clinical Signs and Laboratory Findings
Stage I (Mild disease)	11 (9 M, 3 F)	Positive (8 Lo, 4 Me)	Positive	Onychogryphosis, peripheral lymph-adenomegaly
Stage II (Moderate disease)	9 (4 M, 5 F)	Positive (3 Lo, 6 Me)	Positive	Exfoliative dermatitis, onychogryphosis, ulcerations, epistaxis, non-regenerative anemia, hyperglobulinemia, hypoalbuminemia
Stage III (Severe disease)	7 (4 M, 3 F)	Positive (1 Me, 3 Hi)	Positive	Exfoliative dermatitis, onychogryphosis, ulcerations, epistaxis, weight loss, uveitis and glomerulonephritis, anemia, hyperglobulinemia, hypoalbuminemia, increase of BUN and CREA (1.4–2 mg/dL) [36].
Stage IV (Very severe disease)	3 (2 M, 1 F)	Positive (3 Hi)	Positive	Exfoliative dermatitis, onychogryphosis, thrombocytopenia, ulcerations, epistaxis, weight loss, uveitis, glomerulonephritis, anemia, hyperglobulinemia, hypoalbuminemia, increase of BUN and CREA (>2 mg/dL) [36].

LEISH, dogs infected with *Leishmania* spp.; M, male; F, female; Lo, low titer (≥1:80, ≤1:60); Me, medium titer (≥1:320, ≤1:640); Hi, high titer (≥1: 640 [11]). BUN, Blood urea nitrogen, CREA, Creatinine.

**Table 2 animals-11-00754-t002:** Concentration of HGMB-1 (High Mobility Group Box-1 protein), ROMs (reactive oxidative metabolites), OXY (antioxidant barrier), SHp (thiol groups of plasma compounds) and OSi (ratio between ROMs and OXY) in the serum of dogs affected by CanL (LEISH Group) and in control (CTR Group).

Variable	Unit	LEISH		CTR		*p*-Value
		Mean	SD	Mean	SD	
ROMs	(U CARR)	239.1	103.3	217.3	29.47	Ns
OXY	(μmol HCLO/mL)	409	181	339.3	108.2	Ns
SHp	(μmol/L)	108.7	75.6	65.6	18.3	0.047
OSi	%	57.21	49.54	49.27	20.70	Ns
HMGB-1	(ng/mL)	12.1	10.5	7.3	2.6	Ns

LEISH, dogs infected with *Leishmania* spp.; CTR, dogs uninfected with *Leishmania* spp.; Ns, statistically non-significant differences.

**Table 3 animals-11-00754-t003:** Hematological and biochemical variables in dogs affected by CanL (LEISH Group).

Variable	Unit	LEISH		CTR		Reference Ranges [40,41]	*p*-Value
		Mean	SD	Mean	SD		
RBC	(10^6^/μL)	4.6	1.8	7.3	2.6		< 0.01
Hgb	(ng/mL)	12.7	2.9	15.2	1.99	14.7–17.7	0.02
PCV	(%)	39.3	5.1	43.1	6.2	42–53	0.05
WBC	(103/μL)	14.6	4.9	9.7	1.2	4.6–10.6	< 0.01
PLT	(103/μL)	227	115	290	97	150–400	Ns
UREA	(mg/dL)	40.5	11.5	26.8	11.5	5–21	0.02
CREA	(mg/dL)	1.8	0.5	0.9	0.3	0.3–1.2	< 0.01
ALT	(U/L)	41.6	21.1	35.6	13.2	0–40	Ns
AST	(U/L)	46.3	20.3	34.8	10.7	0–40	Ns
ALB	(g/dL)	2.95	0.46	3.6	0.41	3.0–4.4	< 0.01
GLOB	(g/dL)	3.4	0.40	2.8	0.41	1.8–3.9	0.02
TP	(g/dL)	6.5	0.6	6.4	0.4	6.4–7.9	Ns

RBC, Red blood cell count; PCV, hematocrit; Hgb, hemoglobin concentration; WBC, white blood cell count; PLT, blood platelet count; CREA, Creatinine, ALT, Alanine-amino transferase; AST, Aspartate-amino transferase; ALB, Albumin; GLOB, Globulin; TP, total protein.

**Table 4 animals-11-00754-t004:** Correlation between different HGMB-1 and oxidative stress variables in LEISH Group dogs (Spearman rank order) (* *p* < 0.05).

	ROMs	OXY	SHp	OSi	HMGB-1
ROMs	1.0	−0.30	−0.34	0.77 *	0.40
OXY	−0.30	1.0	−0.35	−0.79 *	−0.13
SHp	−0.34	−0.34	1.0	−0.27	−0.64 *
OSi	0.77 *	−0.79 *	−0.27	1.0	0.42
HMGB-1	0.40	−0.13	−0.64 *	0.42	1.0

ROMs, reactive oxidative metabolites; OXY, antioxidant barrier; SHp, thiol groups of plasma compounds; OSi, ratio between ROMs and OXY; HGMB-1, High Mobility Group Box-1 protein.

**Table 5 animals-11-00754-t005:** Correlation between HMGB-1, oxidative stress and hematochemical analytes in LEISH Group dogs (Spearman rank order) (* *p* < 0.05).

	RBC	Hgb	PCV	WBCs	PLT	ALT	AST	BUN	CREA	ALB	GLOB	TP
ROMs	−0.57 *	0.34	−0.49	0.19	0.15	0.03	0.54	0.13	−0.05	0.21	0.19	−0.31
OXY	0.26	−0.24	0.34	−0.44	0.25	−0.02	−0.30	0.04	0.10	−0.11	−0.28	0.44
SHp	0.21	0.52	0.03	0.15	−0.06	−0.01	0.03	0.08	0.01	−0.23	−0.20	0.34
OSi	0.20	−0.10	−0.45	−0.03	−0.10	0.16	0.30	0.04	0.16	−0.26	0.08	−0.58 *
HMGB-1	0.06	0.26	0.03	−0.07	−0.06	−0.01	0.03	0.08	0.01	0.01	−0.29	0.34

RBC, Red blood cell count; PCV, hematocrit; Hgb, hemoglobin concentration; WBC, white blood cell count; PLT, blood platelet count; BUN, Blood Urea Nitrogen; CREA, Creatinine, ALT, Alanine-amino transferase; AST, Aspartate-amino transferase; ALB, Albumin; GLOB, Globulin; TP, total protein; ROMs, reactive oxidative metabolites; OXY, antioxidant barrier; SHp, thiol groups of plasma compounds; OSi, ratio between ROMs and OXY; HGMB-1, High Mobility Group Box-1 protein.

## Data Availability

The raw data from this research are available upon request to the corresponding author.

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
