# Peer review of "Clinical Significance of ROMs, OXY, SHp and HMGB-1 in Canine Leishmaniosis"

_animals, 2021, doi:10.3390/ani11030754_

Round 1

Reviewer 1 Report

Dear Authors, I find your manuscript "Clinical significance of ROMs, OXY, SHp and HMGB-1 in canine leishmaniosis" very interesting and very well written. In order to be published, your manuscript needs to be corrected - minor revision. Please find attached the corrected version with comments.

Best regards

Author Response

Dear Reviewer 1,

thank you for your suggestion and comments.

Changes were applied as suggested. You can find in MS easily because they are written in red.

Reviewer 2 Report

General comments:

The authors approach an important and over-studied field of investigation, respectively the role of oxidative stress in canine leishmaniasis.

The article is publishable, but it has some shortcomings, as follows:

REVISION

Introduction section

  • line 55: "spp." is the plural form of species abbreviation, indicating "several species". Then, "Leishmania" must be conjugated with the plural form of the following verb, becoming "Leishmania spp. are able...", instead of "is able...".

Material and Methods section

  • line 91: use 40 instead of „forty”.
  • line 93: you can use 30 instead of „thirty”.
  • lines 95-97: use the simplified form: "while the other 10 healthy dogs (7 males and 3 females, with a bodyweight of 14.1 Kg ±2.7, and a mean age 6.4 ± 3.7 years old), serologically, parasitologically and molecularly negative were considered as the control group (Group CTR).".
  • lines 98-101: use the simplified form: "Dogs with associated conditions, such as neoplastic, endocrine, metabolic, infectious (Ehrlichia, Anaplasma spp.), and parasitic (Babesia spp., microfilariae) diseases were excluded from the study.".

Discussion section

  • lines 253-257: Can you hypothesize or explain why the negative correlation between OSi and ROMs, as well as ROMs and RBC, was detected in the LEISH group when usually a positive correlation would be expected?
  • I think several other papers aiming the same topic, the role of oxidative stress in CanL should be referenced and discussed in your manuscript, even if those authors approached other markers of this stress, namely:
    • Almeida, B. F. M., Narciso, L. G., Melo, L. M., Preve, P. P., Bosco, A. M., Lima, V. M. F., & Ciarlini, P. C. (2013). Leishmaniasis causes oxidative stress and alteration of oxidative metabolism and viability of neutrophils in dogs. The Veterinary Journal, 198(3), 599–605. doi:10.1016/j.tvjl.2013.08.024
    • Solcà, M., Andrade, B., Abbehusen, M. et al. Circulating Biomarkers of Immune Activation, Oxidative Stress and Inflammation Characterize Severe Canine Visceral Leishmaniasis. Sci Rep 6, 32619 (2016). https://doi.org/10.1038/srep32619
    • Quintavalla, F.; Basini, G.; Bussolati, S.; Carrozzo, G.G.; Inglese, A.; Ramoni, R. Redox Status in Canine Leishmaniasis. Animals 2021, 11, 119. https://doi.org/10.3390/ani11010119

Author Response

Dear Reviewer 2,

thank you for your suggestion and comments.

Changes were applied as suggested. You can find in MS easily because they are written in red.

REVISION

Introduction section

  • line 55: "spp." is the plural form of species abbreviation, indicating "several species". Then, "Leishmania" must be conjugated with the plural form of the following verb, becoming "Leishmania spp. are able...", instead of "is able...".

The sentence has been modified.

Material and Methods section

  • line 91: use 40 instead of „forty”.

Forty has been modified in 40.

  • line 93: you can use 30 instead of „thirty”.

Thirty has been modified in 30.

  • lines 95-97: use the simplified form: "while the other 10 healthy dogs (7 males and 3 females, with a bodyweight of 14.1 Kg ±2.7, and a mean age 6.4 ± 3.7 years old), serologically, parasitologically and molecularly negative were considered as the control group (Group CTR)."

The sentence has been modified as suggested.

  • lines 98-101: use the simplified form: "Dogs with associated conditions, such as neoplastic, endocrine, metabolic, infectious (EhrlichiaAnaplasma spp.), and parasitic (Babesia spp., microfilariae) diseases were excluded from the study."

The sentence has been modified as suggested.

Discussion section

  • lines 253-257: Can you hypothesize or explain why the negative correlation between OSi and ROMs, as well as ROMs and RBC, was detected in the LEISH group when usually a positive correlation would be expected?

The type error was present. The error has been corrected.

  • I think several other papers aiming the same topic, the role of oxidative stress in CanL should be referenced and discussed in your manuscript, even if those authors approached other markers of this stress, namely:
    • Almeida, B. F. M., Narciso, L. G., Melo, L. M., Preve, P. P., Bosco, A. M., Lima, V. M. F., & Ciarlini, P. C. (2013). Leishmaniasis causes oxidative stress and alteration of oxidative metabolism and viability of neutrophils in dogs. The Veterinary Journal, 198(3), 599–605. doi:10.1016/j.tvjl.2013.08.024
    • Solcà, M., Andrade, B., Abbehusen, M. et al. Circulating Biomarkers of Immune Activation, Oxidative Stress and Inflammation Characterize Severe Canine Visceral Leishmaniasis. Sci Rep 6, 32619 (2016). https://doi.org/10.1038/srep32619
    • Quintavalla, F.; Basini, G.; Bussolati, S.; Carrozzo, G.G.; Inglese, A.; Ramoni, R. Redox Status in Canine Leishmaniasis. Animals 2021, 11, 119. https://doi.org/10.3390/ani11010119.

The references suggested have been included and discussed.

Reviewer 3 Report

The study conducted by Pugliese and collaborates is very important in the field. The results of the paper will improve the knowledge about the mechanisms in leishmaniasis. Whilst looking forward to reading this paper, I was ultimately somewhat disappointed. Partly this was down to a poor quality of english, but also since I felt an opportunity was missed. 

Line 12: change ‘oxidative reactive species” into “reactive oxidative metabolites’

Line 43. The comma should be before “and” not after

Line 53. Change “unbalance” into “imbalance”

Line 56. Change “trough” into “through”

Line 60. Don’t understand why authors write about oxidative stress in haemolytic anaemia, when the introduction should be about leishmaniasis. Why didn’t the authors citate in the introduction the study conducted by Kumar et al. [2017] concerning the oxidative stress parameters in the kidneys of mice experimentally infected with Leishmania donovani? It is more suitable to the topic of paper than haemolytic anaemia.

Kumar V, Tiwari N, Gedda MR, Haque R, Singh RK. Leishmania donovani infection activates Toll-like receptor 2, 4 expressions and Transforming growth factor-beta mediated apoptosis in renal tissues. Braz J Infect Dis. 2017;21(5):545-549. doi: 10.1016/j.bjid.2017.04.007.

Line 63. I would suggest to put the paragraph (line 63-65) after paragraph found in the line 67-71

Line 72. I don’t understand what authors wanted to write in the paragraph starting in the line 72.

Line 75. Change “oxidative reactive species” into “reactive oxidative metabolites’

Line 77. In what biological samples of dogs, authors wanted to examine parameters of oxidative stress?

Line 78. Put the full stop after Leishmania spp. and write “Moreover, the objective was to investigate the relationship between ….”

Line 79. Delete “suggestive”

Line 94 and 96: write kg in small letters

Line 98.  The sentence is not in English. Please write “Dogs with the concomitant neoplastic, endocrine, metabolic or infective diseases were excluded from the study”

Line 99. spp. shouldn’t be in italics

Line 100/101: rewrite the sentence

Line 105: delete “on biological tissue”

Lines 106-113: rewrite the paragraph

Line 115: what does it mean that PCR was performed on whole blood and/or (?) lymph node samples? Why all dogs didn’t have the same test?

Line 139. The results of OXY are expressed as…?

Line 140: please write “derivatives of reactive oxygen metabolites” before d-ROMS

Line 149. Please write “in the” before serum samples

Lines 153-158. You didn’t write which heamatological parameters were examined.

Material and methods: Did you performed all the assays in duplicate samples?

Line 170. Put the word “group” after “LEISH”

Line 177. Put the information which are presented under the tables (line 178 and 179) in the title of the table in the brackets.

Line 183. Add “compared to control group” after “higher value of SHp”

Line 188. Delete the sentence “the values of oxidative biomarkers …”

Line 191: Which two groups do you have in mind? Group I and II or LEISH and CTR?

Line 194. Describe the abbreviations used in the tables in the title of of the table in the brackets.
 Table 2. Concentration of HGMB-1, ROMs, OXY and SHp in the serum or plasma of dogs (LEISH, dogs infected with Leishmania spp.; CTR, dogs uninfected with Leishmania spp.; NS, statistically non-significant differences)

Line 203. Comma before “and” not after

Line 203. Don’t understand the sentence “no correlation was found between haematological parameters and indices of liver and kidney injuries” please rewrite. Besides, based on which parameters authors define liver and kidney injuries??

Line 210: please write “haematological and biochemical analytes” instead of “haematochemical”

Lines 216-230. It is more introduction than discussion. In the line 228 Leishmania should be in italics

Line 237. Write “H2O2” instead of “H2O2”

Discussion. Please add more interpretation of your results.

Author Response

Dear Reviewer 3,

thank you for your suggestion and comments.

Changes were applied as suggested. You can find in MS easily because they are written in red.

REVISION

Line 12: change ‘oxidative reactive species” into “reactive oxidative metabolites’.

The sentence has been modified as suggested.

Line 43. The comma should be before “and” not after.

The comma has been inserted before “and”.

Line 53. Change “unbalance” into “imbalance”.

The sentence has been modified as suggested.

Line 56. Change “trough” into “through”.

The word has been corrected.

Line 60. Don’t understand why authors write about oxidative stress in haemolytic anaemia, when the introduction should be about leishmaniasis. Why didn’t the authors citate in the introduction the study conducted by Kumar et al. [2017] concerning the oxidative stress parameters in the kidneys of mice experimentally infected with Leishmania donovani? It is more suitable to the topic of paper than haemolytic anaemia.

Kumar V, Tiwari N, Gedda MR, Haque R, Singh RK. Leishmania donovani infection activates Toll-like receptor 2, 4 expressions and Transforming growth factor-beta mediated apoptosis in renal tissues. Braz J Infect Dis. 2017;21(5):545-549. doi: 10.1016/j.bjid.2017.04.007.

The sentence has been modified and the reference has been added.

Line 63. I would suggest to put the paragraph (line 63-65) after paragraph found in the line 67-71.

Paragraph has been modified as suggest.

Line 72. I don’t understand what authors wanted to write in the paragraph starting in the line 72.

The sentence has been modified.

Line 75. Change “oxidative reactive species” into “reactive oxidative metabolites’

The sentence has been modified as suggested.

Line 77. In what biological samples of dogs, authors wanted to examine parameters of oxidative stress?

The parameters of oxidative stress have been assayed on blood samples of dogs infected with Leishmania.

The information has been added at the paragraph.

Line 78. Put the full stop after Leishmania spp. and write “Moreover, the objective was to investigate the relationship between ….”

The sentence has been modified as suggested.

Line 79. Delete “suggestive”.

The word has been deleted.

Line 94 and 96: write kg in small letters

Kg has been written in small letters.

Line 98.  The sentence is not in English. Please write “Dogs with the concomitant neoplastic, endocrine, metabolic or infective diseases were excluded from the study”.

The sentence has been modified also in accordance to Rev 2.

Line 99. spp. shouldn’t be in italics.

Spp. has been modified in italics.

Line 100/101: rewrite the sentence.

In accordance to the reviewer 2 the paragraph has been modified.

Line 105: delete “on biological tissue”.

Biological tissue has been deleted.

Lines 106-113: rewrite the paragraph.

The paragraph has been rewritten.

Line 115: what does it mean that PCR was performed on whole blood and/or (?) lymph node samples? Why all dogs didn’t have the same test?

The sentence has been correct.

Line 139. The results of OXY are expressed as…?

OXY values are expressed in mmol/L. The information has been added at the paragraph.

Line 140: please write “derivatives of reactive oxygen metabolites” before d-ROMS

The sentence has been modified as suggested.

Line 149. Please write “in the” before serum samples.

“In the” has been added before serum samples.

Lines 153-158. You didn’t write which heamatological parameters were examined.

Heamatological parameters examined were included in the paragraph.

Material and methods: Did you performed all the assays in duplicate samples?

All samples were assayed in duplicate.

Line 170. Put the word “group” after “LEISH”.

The word has been put in the correct place.

Line 177. Put the information which are presented under the tables (line 178 and 179) in the title of the table in the brackets.

The information is reported in the title of thetable.

Line 183. Add “compared to control group” after “higher value of SHp”

The sentence has been modified.

Line 188. Delete the sentence “the values of oxidative biomarkers …”

The sentence has been deleted.

Line 191: Which two groups do you have in mind? Group I and II or LEISH and CTR?

The sentence has referred to LEISH and CTR.

Line 194. Describe the abbreviations used in the tables in the title of of the table in the brackets.
 Table 2. Concentration of HGMB-1, ROMs, OXY and SHp in the serum or plasma of dogs (LEISH, dogs infected with Leishmania spp.; CTR, dogs uninfected with Leishmania spp.; NS, statistically non-significant differences).

Abbreviations have been described in the title of the table.

Line 203. Comma before “and” not after.

Comma has been put before “and”.

Line 203. Don’t understand the sentence “no correlation was found between haematological parameters and indices of liver and kidney injuries” please rewrite. Besides, based on which parameters authors define liver and kidney injuries??

The sentence has been modified.

Line 210: please write “haematological and biochemical analytes” instead of “haematochemical”.

“Haematological and biochemical analytes” have been modified in “haematochemical”.

Lines 216-230. It is more introduction than discussion. In the line 228 Leishmania should be in italics.

In this paragraph, the authors introduced the discussion of the results. Leishmania has written in italics.

Line 237. Write “H2O2” instead of “H2O2”.

The word has been changed as suggested

Discussion. Please add more interpretation of your results.

The discussion has been improved, detailing the results.

Round 2

Reviewer 1 Report

Dear Authors, according to all improved changes my recommendation is to  accept your manuscript.

Author Response

The Authors thanks the reviewer

Reviewer 3 Report

After the authors' corrections, I strongly recommend publishing the paper.

Author Response

The Authors thanks the reviewer